# Weakening of the Tropical Tropopause Layer Cold Trap with Global Warming

Stephen Bourguet[1] and Marianna Linz[1, 2]

[1]Harvard University Department of Earth and Planetary Sciences, 20 Oxford Street, Cambridge, MA 02138
[2]Harvard University School of Engineering and Applied Sciences, 29 Oxford Street, Cambridge, MA 02138

**Correspondence:** Stephen Bourguet (stephen_bourguet@g.harvard.edu)

**Abstract.** Lagrangian trajectories have previously been used to reconstruct water vapor variability in the lower stratosphere, where the sensitivity of surface radiation to changes in the water vapor concentration is strongest, by obtaining temperature histories of air parcels that ascend from the troposphere to the stratosphere through the tropical tropopause layer (TTL). Models and theory predict an acceleration of the Brewer-Dobson Circulation (BDC) and deceleration of the Walker Circulation with surface warming, and both of these will drive future changes to transport across the TTL. Here, we examine the response of TTL transport during boreal winter to idealized changes in the BDC and Walker Circulation by comparing the temperature histories of trajectories computed with ERA5 data to those calculated using the same data but with altered vertical and zonal wind velocities. We find that lower stratospheric water vapor mixing ratios calculated from trajectories' cold point temperatures can increase by about 1.6 ppmv (about 50%) when only zonal winds are slowed, while changes to vertical winds have a negligible impact on water vapor concentrations. This change follows from a decrease in zonal sampling of the temperature field by trajectories, which weakens the "cold trap" mechanism of dehydration as TTL transport evolves. As the zonal winds of the TTL decrease, the fraction of air that passes through the cold trap while ascending to the stratosphere will decrease and the coldest average temperature experienced by parcels will increase. Future changes to TTL temperatures can be applied as an offset to these temperature histories, including enhanced warming of the cold trap due to "El Niño"-like warming, which has a secondary impact on the fraction of air that is dehydrated by the cold trap. Some of the resultant moistening may be negated by a decreased rate of temperature change following the cold point, which will allow more ice to gravitationally settle before sublimating outside of the cold trap. This result presents a mechanism for a stratospheric water vapor feedback that can exist without changes to TTL temperatures.

## 1 Introduction

The cold temperatures of the tropical tropopause layer (TTL) dehydrate air as it ascends from the troposphere into the lower stratosphere, resulting in extreme dryness throughout the middle atmosphere. In 1949, observations of low relative humidity above the mid-latitude tropopause led Alan Brewer to deduce the equator-to-pole overturning circulation of the stratosphere (Brewer, 1949), now known as the Brewer–Dobson circulation (BDC). The seasonal and interannual variability of water vapor concentrations in the lower stratosphere can also be explained by temperature fluctuations in the TTL, and the ascent of

these anomalies up into the stratosphere give an indirect line of evidence for the TTL acting as a "gateway" for air entering the stratosphere, as well as the strength of the BDC (the "water vapor tape recorder"; Mote et al. (1996); Randel and Park (2019)). Surface climate is most sensitive to changes to water vapor concentrations near the tropopause (Forster and Shine, 1999; Solomon et al., 2010; Riese et al., 2012), which are predicted to be increase with surface warming and act as a positive feedback on greenhouse gas forcing (Dessler et al., 2013; Gettelman et al., 2010; Keeble et al., 2021).

Although an Eulerian, zonal-mean view of TTL temperatures can approximate the dryness of the stratosphere, a Lagrangian perspective is required to understand the importance of four–dimensional temperature variability in the dehydration process (Fueglistaler et al., 2005). Water vapor reconstructions using Clausius-Clapeyron scaling at the coldest temperatures experienced by trajectories (i.e. the Lagrangian cold point) can reproduce interannual variability in lower stratospheric mixing ratios and have been used to infer a multidecadal moistening of the stratosphere (Smith et al., 2021; Konopka et al., 2022), yet these reconstructions are drier than observations (Liu et al., 2010). Therefore, some combination of other difficult-to-quantify processes (e.g. ice nucleation and sublimation, supersaturation, overshoot convection) must be at play for the observed quantity of stratospheric water vapor to persist (Dessler et al., 2007; Corti et al., 2008; Ueyama et al., 2015, 2020). We will not directly study these processes, but they will remain important when considering ice formation and sublimation in the context of the temperature history of air parcels passing through the TTL.

Work with Lagrangian trajectories supports the "cold trap" hypothesis, which invokes the relative magnitudes of zonal and vertical flow within the TTL to suggest that air parcels travel great horizontal distances within the TTL, allowing them to experience the TTL's coldest temperatures regardless of where they enter or exit the TTL (Holton and Gettelman, 2001; Pan et al., 2019). This is crucial for explaining the dehydration of air entering the stratosphere, and it reconciles the incompatibility of the previous "stratospheric fountain" hypothesis with observations of net subsidence in the TTL's coldest regions (Newell and Gould-Stewart, 1981; Sherwood, 2000). The TTL winds are driven horizontally by the uppermost flow of the troposphere's Walker Circulation and vertically by the shallow branch of the BDC (Fueglistaler et al., 2009). Therefore, it is reasonable to expect the power of the cold trap hypothesis to depend on the vertical and zonal winds within the TTL, and thus the strength of the BDC and Walker Circulation, but this has not been systematically analyzed before.

There has been extensive work on future changes to the BDC and the Walker Circulation outside of the context of dehydration within the TTL. An acceleration of the vertical velocity at 70 hPa is a robust response to greenhouse gas forcing in climate models (Butchart, 2014), although this has also been interpreted as an upward shift of the BDC, and surface warming may only drive changes to the BDC's shallow branch (Oberländer-Hayn et al., 2016; Abalos et al., 2021). Conversely, thermodynamic constraints imposed by the hydrological cycle dictate a decrease in strength for the Walker Circulation (Held and Soden, 2006), yet recent observations of Walker Circulation strength are equivocal and tropical Pacific SSTs can drive a temporary Walker Circulation acceleration (Lee et al., 2022; Chung et al., 2019; Heede et al., 2021). Nonetheless, Walker Circulation weakening is a robust response to warming in models (Vecchi and Soden, 2007).

Following the projections of increased BDC strength, Fueglistaler et al. (2014) explored how changes to the BDC induce changes to both TTL temperatures and transport. Using Lagrangian trajectories, they showed that the response of the lower stratospheric water vapor mixing ratio to increased dynamic cooling (an accelerated BDC) deviates from the response expected

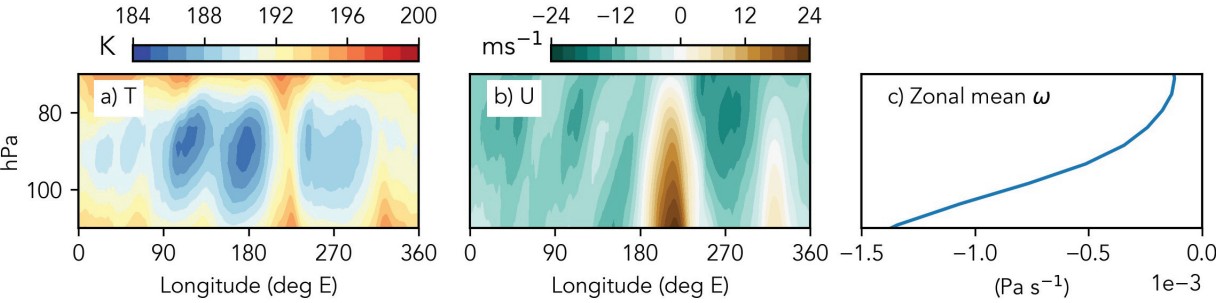

**Figure 1.** Meridional mean (10° S to 10° N) ERA5 tropical tropopause layer (a) temperatures, (b) zonal winds, and zonal mean vertical velocity for January 20–29, 2008. Note that positive zonal wind is eastward and negative vertical wind is upward.

from Clausius-Clapeyron scaling by about 10% due to an decrease in the sampling of spatial and temporal temperature variability within the TTL. This means that some of the dryness induced by cooling following an acceleration of the BDC will be offset by changes to transport, which will combine with increased TTL temperatures to increase the humidity of the lower stratosphere. Changes to the zonal winds were not explored though, leaving open the possibility of further deviations from Clausius-Clapeyron scaling.

Here, we revisit the impact of future changes to transport across the TTL on the Lagrangian cold point by considering changes to both the zonal and vertical wind speeds and explore their impacts on lower stratospheric water vapor. We do so using Lagrangian trajectories run with ERA5 wind fields altered based on future changes projected by models and observations. In section 2, we describe our methods and data; in section 3, we show trajectory results and calculate their differential impacts on water vapor; and in section 4, we summarize and discuss implications of this study.

## 2 Methods

### 2.1 Model and data

We use ECMWF's Lagrangian analysis tool LAGRANTO version 2 (Sprenger and Wernli, 2015) run with ERA5 (ECMWF's latest reanalysis) input data (Hersbach et al., 2020). Figure 1 shows the relevant meteorological variables within the region of study from January 20–29, 2008, which is when many trajectories analyzed here experience their cold point. The cold region west of the date line (i.e. the cold trap) is a striking feature of the temperature field, while the divergent flow of the Walker Circulation dominates the zonal wind field.

Following the procedure of Bourguet and Linz (2022), we use 0.5° x 0.5° horizontal resolution data on 137 vertical layers with 1-h temporal resolution. The vertical winds are kinematic, not diabatic (i.e. pressure tendencies rather than heating rates), which has been shown to cause excess dispersion for trajectories run with lower resolution data, but is less impactful for input data with ERA5's resolution (Liu et al., 2010; Legras and Bucci, 2020; Bourguet and Linz, 2022). We initialize trajectories on

a 0.5° x 0.5° grid at all longitudes from 20° S to 20° N at the end of February and run them backwards to the beginning of December of the previous year. Four sets of trajectories separated by six hours are run per year for a total of over 200,000 per experiment. The starting heights are interpolated to the pressure levels corresponding to the 400 K isentrope, but the trajectories are run on pressure levels and are driven by the pressure velocity. See Bourguet and Linz (2022) for further discussion and justification of this method.

## 2.2 Modified transport experiments

To test the response of trajectory temperature histories to changes in TTL transport, we run trajectories with wind fields modified based on idealized changes to the BDC and Walker Circulation. Experiments are done during boreal winter (DJF), when transport across the TTL is strongest (Rosenlof, 1995), with the year corresponding to the JF year. Trajectories are run for 2007, 2008, and 2009, and the results presented here are from 2008. Figures for 2007 and 2009 corresponding to those in Section 3 and tables corresponding to Table 2 are provided in the supplemental and appendix, respectively, and show that the results for 2008 are robust. These experiments are summarized in Table 1.

We chose to only test an accelerated BDC in this work because there is no indication that its strength could decrease with surface warming, and we test both an accelerated and decelerated Walker Circulation based on the evidence of a recent strengthening and the expected weakening from models and theory. An accelerated BDC would also increase dynamic cooling within the TTL, but as we show in Section 2.3, increased greenhouse gas concentrations have a net warming effect in this layer. We aim to isolate the impacts of transport in this study, so we do not impose a temperature change.

The range of vertical wind velocities considered here reflect changes projected by CMIP5, CMIP6, and other modeling studies with increased $CO_2$ mixing ratios (Butchart et al., 2010; Hardiman et al., 2014; Chrysanthou et al., 2020; Abalos et al., 2021). The 20% zonal wind acceleration is justified by the the observed 14% strengthening of the Walker Circulation circulation in the final three decades of the last century as calculated using the water vapor flux in the lower branch of the circulation (Sohn and Park, 2010), and 20% deceleration is justified by the prediction of a weakening of about 5% $^oC^{-1}$ with future warming from theory (Held and Soden, 2006). The 50% zonal wind acceleration and declaration scenarios are extreme, but they are nonetheless useful for considering how any change in zonal winds could impact trajectory temperature histories and the dehydration process.

We accelerate the BDC by applying a constant offset equal to a fraction of the zonal mean to the vertical winds everywhere (i.e. $\omega = \omega + a\overline{\omega}$, where a is the fractional change and $\overline{\omega}$ is the zonal mean upwelling rate at 100 hPa of -0.0007 Pa $s^{-1}$). Because the vertical wind strength decreases with height within the TTL (see Fig. 1c), this offset overestimates the fractional increase to the zonal mean upwelling above 100 hPa. As is discussed in Section 3.3, this approach helps ensure that vertical winds are accelerating appreciably in regions where the upwelling is anomalously strong. A scaled approach (i.e. $\omega = \omega + a\omega$) would resolve this issue, but it would also increase dispersion by amplifying variability in the vertical winds associated with local waves and fail to isolate a strengthening BDC. The results of Fueglistaler et al. (2014) were not sensitive to the choice of scaled or offset vertical winds, so we feel that this approach is best when altering vertical winds by as much as 50%.

| Scenario | Horizontal wind scaling | Vertical wind scaling |
|---|---|---|
| Modern | 1 | 1 |
| Moderate BDC increase | 1 | 1.2 |
| Strong BDC increase | 1 | 1.5 |
| Moderate Walker Circulation increase | 1.2 | 1 |
| Moderate WC increase + Moderate BDC increase | 1.2 | 1.2 |
| Strong Walker Circulation increase | 1.5 | 1 |
| Strong WC increase + Strong BDC increase | 1.5 | 1.5 |
| Moderate Walker Circulation decrease | 0.8 | 1 |
| Moderate WC decrease + Moderate BDC increase | 0.8 | 1.2 |
| Strong Walker Circulation decrease | 0.5 | 1 |
| Strong WC decrease + Strong BDC increase | 0.5 | 1.5 |

**Table 1.** Summary of the LAGRANTO runs used to test the sensitivity of trajectory temperature histories to circulation scenarios (WC = Walker Circulation). Runs were done for DJF 2007, 2008, and 2009. Results shown are from DJF 2008, while DJF 2007 and 2009 results can be found in the SI.

On the other hand, the zonal winds within the TTL are divergent (see Fig. 1b), and future changes that follow the strength of
115 the Walker Circulation will amplify or dampen this pattern. Therefore, it is necessary to apply a scaling factor to the zonal wind field, rather than an offset (i.e. U = aU). This results in an unphysical wind field with regions of zonal divergence or convergence that are not compensated for by vertical or meridional flux, which could lead trajectories to cross the cold trap at the incorrect latitude or reside in the cold trap for the wrong period of time. If the cold trap had a very narrow meridional extent, then this pattern could bias the trajectories' temperature sampling by exposing them to the incorrect temperature region. The cold trap
over the West Pacific spans approximately 15° S to 15° N (Fueglistaler et al., 2005; Bourguet and Linz, 2022), so we argue that trajectories will largely sample the same cold temperatures regardless of whether the meridional winds are appropriately scaled based on the meridional broadness of the cold trap. In other words, the meridional distribution of trajectories within the TTL may not be consistent with what it would be if the meridional winds were scaled properly, but the relative uniformity of temperatures across latitudes allows us to disregard this potential sampling issue. (Sample spatial distributions of DJF 2008
trajectories with varied vertical and zonal winds after 40 days of integration are shown in Fig. S1. The exact distribution patterns differ between these plots, which is expected given the different wind fields, and the trajectories only appear to cluster together when zonal winds are decreased by 50%. While a decrease in the meridional winds would likely decrease this effect by spreading these clustered trajectories across nearby latitudes, this is not problematic for our results because of the previously mentioned meridional extent of the cold trap.)
This scaled approach also creates excess dispersion in the accelerated zonal wind scenarios, which causes trajectories to undersample the spatial and temporal variability of the temperature field. This effect does not impact our conclusions regarding the decelerated Walker Circulation, though it does obscure the decreased mean cold point temperature that results from the

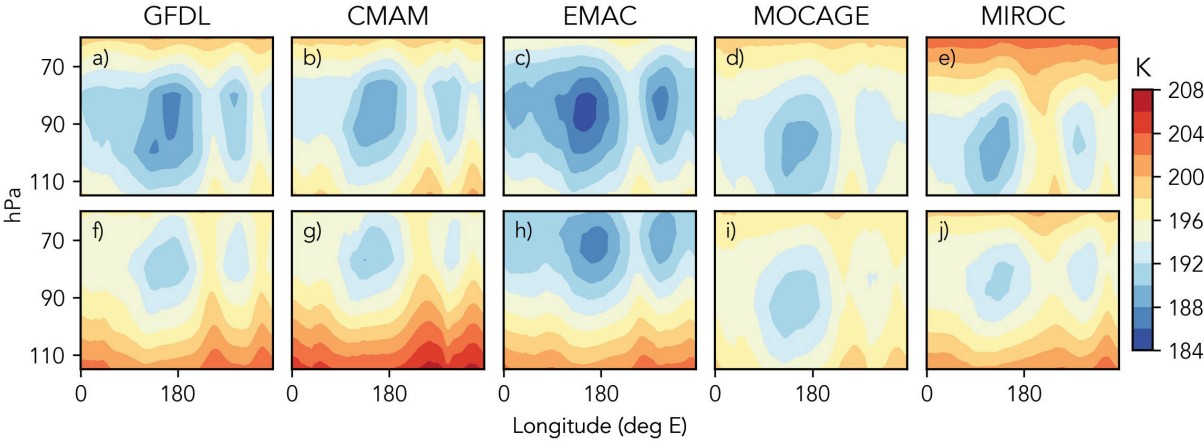

**Figure 2.** Meridional mean tropical tropopause layer January temperatures for 2000-2009 (top row) and 2100-2109 (bottom row) from historical and RCP 8.5 runs from the Atmospheric Chemistry and Climate Model Intercomparison (ACCMIP; Lamarque et al. (2013)). Models shown are a/e) GFDL-AM3 (Donner et al., 2011), b/g) CMAM (Scinocca et al., 2008), c/h) EMAC (Jöckel et al., 2016), d/i) MOCAGE (Teyssèdre et al., 2007), and e/j) MIROC-CHEM (Watanabe et al., 2011). Annual mean temperatures are shown in Fig. S2.

accelerated Walker Circulation, especially when zonal winds are increased by 50%. This dispersion decreases the amount of time spent in the TTL and the spatial sampling therein, so it introduces a warm bias that counteracts the decrease in the cold point temperature associated with trajectories traversing greater zonal distances. Thus, our results for the accelerated zonal wind trajectories underestimate the effect of increased temperature sampling, but we do not attempt to correct for this given our focus on the decelerated zonal wind scenarios.

### 2.3 Zonal mean warming assumption

Our method is sensitive to the temperature structure of the TTL and patterns of future warming within it, so we feel that it is necessary to justify ignoring future changes to the TTL's zonal temperature structure and note this assumption as a caveat to our results. If the cold point were to warm more quickly than the rest of the TTL, then it would no longer be a cold trap: zonal transport would become less important and the trajectories' cold point longitudes would spread uniformly across the TTL. Conversely, if the cold trap were to not warm while the rest of the TTL warmed, then lower stratospheric water vapor concentrations would become more sensitive to changes in transport. By only modifying the winds, we assume that Lagrangian trajectory analyses of future warming are only impacted by changes in transport, while temperature changes can be applied as an offset to all trajectories.

Figure 2 shows the meridional mean January temperatures for 2000–2009 and 2100–2109 (other than MOCAGE, for which only 2000–2003 and 2100–2103 are available) from historical and RCP 8.5 runs from the Atmospheric Chemistry and Climate Model Intercomparison (ACCMIP; Lamarque et al. (2013)). Annual mean temperatures can be found in Fig. S2. Qualitatively,

there is not a clear zonal pattern to warming within the TTL in any of the models. Instead, the temperature structure appears to move upwards while warming at all longitudes, which is consistent with tropospheric expansion and increased local radiative heating in the TTL (Lin et al., 2017).

Quantitatively, there is a small amplification of warming in the cold trap, with increases of 1.7 K to the minimum zonal mean temperature and 2.3 K of the minimum mean between 120° E and 180° E (these numbers are 1.7 K and 2.1 K for the models' annual mean). This amplified warming within the cold trap will weaken the relevance of the cold trap hypothesis, but it does not eliminate the cold trap: the temperatures between 120° E and 180° E remain about 5 K colder than the zonal mean temperature. This amplification of cold trap warming has previously been linked to the deceleration of the Walker Circulation (Hu et al., 2016), so our altered zonal wind scenarios would likely be impacted by patterns of temperature change similar to those shown in Fig. 2. To test this, we add 0.6 K to the temperature field between 120° E and 180° E for trajectories run with modern winds and with zonal winds reduced by 20% in 2008. By accounting for this asymmetric warming, the percent of trajectories that experience their cold point in this region from 57% to 51% and 46% to 41%, respectively. Thus, changes to the zonal wind speed dominate the response of the cold trap hypothesis to surface warming and our method is justified.

## 3    Results

### 3.1    Cold point temperature

Figure 3 shows the impact of altered transport on the coldest temperatures experienced by trajectories transiting the TTL. By comparing the left and right columns (decelerated and accelerated zonal winds, respectively), we can see that changes to the zonal winds drive changes to the Lagrangian cold point temperature. This is most obvious for the 50% change scenarios: the orange line (decelerated zonal winds) in panel c is shifted to the right of the black line (unaltered winds), with an increase in mean temperature from 183.4 K to 186.7 K. Conversely, the distribution for the accelerated zonal wind trajectories (orange line) in panel d is shifted slightly to the left of the black line, and the corresponding mean cold point temperatures decreases to 183.1 K. The trajectories with 50% zonal wind acceleration are impacted by excess dispersion, so this temperature decrease could be even greater without unphysical outliers.

This effect of decreased zonal wind speeds on trajectories' cold point temperatures is smaller in 2007 and 2009, with increased mean cold point temperatures of 1.9 K and 2.1 K for trajectories with zonal wind speeds decelerated by 50% in those years, respectively (Figs. S3 and S4). The variability across these years can likely be explained by the El Niño/Southern Oscillation (ENSO) state, which was weakly positive in 2007, weakly negative in 2008, and transitioning in early 2009. A positive ENSO index corresponds to a shifted and more diffuse cold trap, so changes to the zonal winds in the TTL should have less of an impact during El Niño years. This is an analog for the future warming pattern described in Section 2.3 – El Niño-like warming may lessen but not eliminate the importance of zonal winds for TTL dehydration.

Although a 50% change in the TTL's zonal winds is extreme, this pattern of change also emerges in the 20% scenarios. The decelerated zonal wind trajectories' mean cold point temperature in panel a rises from 183.4 K to 184.2 K, indicating that the average coldest temperature experienced by air parcels ascending into the tropical lower stratosphere would increase if the

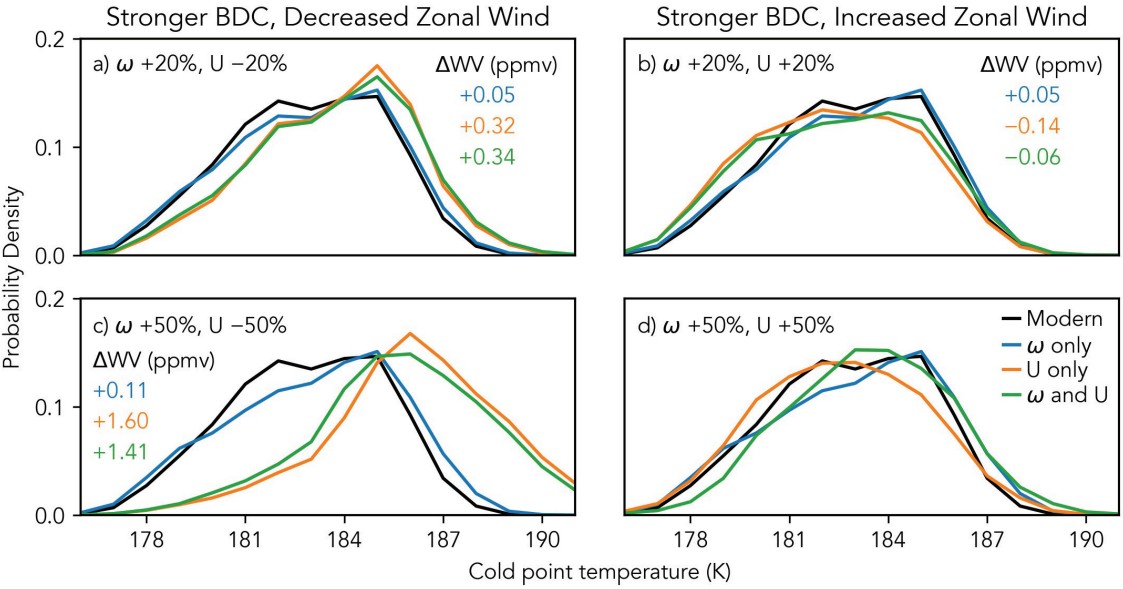

**Figure 3.** Probability density functions of trajectory cold point temperatures for DJF 2008 for the four altered transport scenarios: a) BDC accelerated by 20% and Walker Circulation decelerated by 20%, b) BDC accelerated by 20% and Walker Circulation accelerated by 20%, c) BDC accelerated by 50% and Walker Circulation decelerated by 50%, d) BDC accelerated by 50% and Walker Circulation accelerated by 50%. The corresponding increase in water vapor based on Clausius-Clapeyron scaling is indicated in panels a, b, and c; we omit these values in panel d due to the impact of dispersion on those trajectories. Figs. S3 and S4 show plots for DJF 2007 and 2009.

Walker Circulation were to decelerate as theory predicts. (These changes are smaller again for 2007 and 2009: 0.2 K and 0.3 K, respectively.) For the observed recent Walker Circulation strengthening, the mean cold point temperature for air transiting the

TTL may have decreased, as is indicated by the mean cold point temperature of 182.8 K for the trajectories with zonal winds accelerated by 20% shown in panel b. The impacts of the temperature shifts on saturation water vapor mixing ratios are noted in panels a, b, and c and will be discussed further in Section 3.4. We do not include the saturation water vapor mixing ratios in panel d because the increased dispersion of the accelerated trajectories masks the decrease in the mean cold point temperature.

    The distributions of cold points measured with accelerated vertical winds have a small amount of weight shifted to their

warm tails, but this effect does not change the mean cold point temperature for the trajectories accelerated by 20% and it only changes the mean cold point temperature from 183.4 K to 183.6 K for the trajectories accelerated by 50%. This change is larger for 2007 and 2009 (0.5 K and 0.3 K, respectively), but these results differ nonetheless from that of Fueglistaler et al. (2014), who found that a 50% increase in the BDC strength resulted in an increase of about 0.8 K for the trajectory cold point temperature. We do not necessarily refute their findings though. The experiments run here were only for DJF, which is

when Fueglistaler et al. (2014) found that an accelerated BDC had the smallest impact on transport induced warming to the

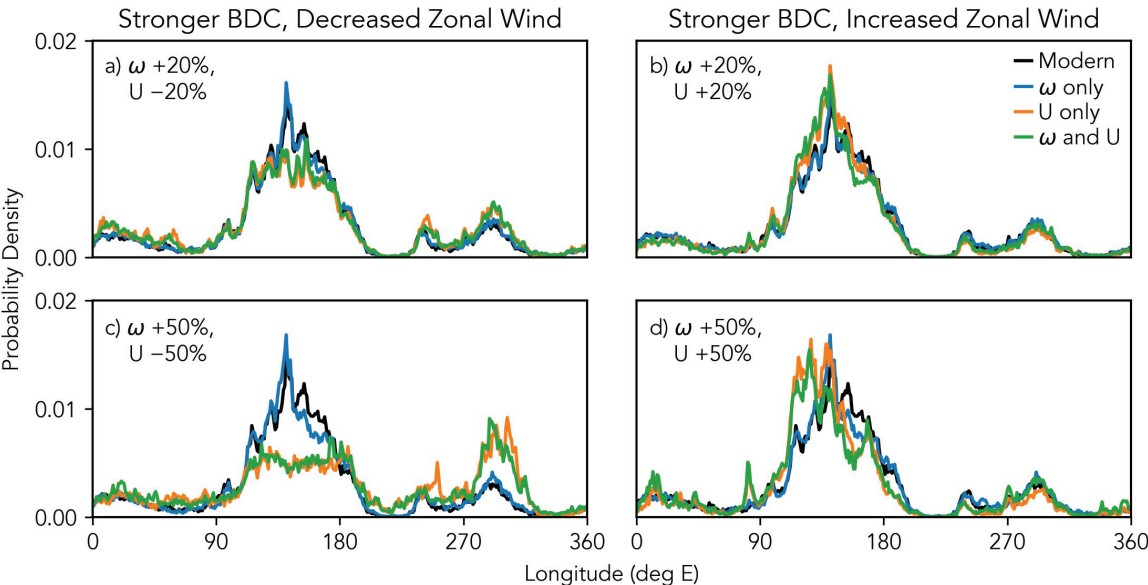

**Figure 4.** Probability density functions of trajectory cold point longitudes for DJF 2008 for the four altered transport scenarios: a) BDC accelerated by 20% and Walker Circulation decelerated by 20%, b) BDC accelerated by 20% and Walker Circulation accelerated by 20%, c) BDC accelerated by 50% and Walker Circulation decelerated by 50%, d) BDC accelerated by 50% and Walker Circulation accelerated by 50%. Figs. S5 and S6 show plots for DJF 2007 and 2009.

Lagrangian cold point, so it is possible that this change could be larger for trajectories run throughout the year. Regardless, the change in cold point temperature driven by the modified vertical wind shown here is a fraction of the modified zonal wind.

### 3.2 Cold point longitude

Figure 4 shows the longitude of the trajectories' cold points, which illustrates the underlying mechanism for warming induced by zonal wind changes. In short, the strength of the cold trap hypothesis depends on the strength of the zonal winds: when the winds are weakened, less air passes through the cold trap between 120° E and 180° E, which allows air to transit the TTL without experiencing the uniquely cold temperatures within.

This impact is again most obvious when comparing panels c and d. In c, when the zonal wind is reduced by 50%, the percent of trajectories experiencing their cold point between 120° E and 180° E decreases from 57% to 30%, with the majority of these warmer cold points experienced between about 270° E and 330° E. In 2009, the percent of trajectories experiencing their cold point between 120° E and 180° E decreases from 52% to 27% when zonal winds are decreased by 50%, while in 2007 the decrease is from 30% to 23% (Figs. S5 and S6). As was the case with the cold point temperatures, this is likely a signature of ENSO variability: the shifted and dispersed cold trap during the weak El Niño year (2007) causes fewer trajectories to experience their cold point within the region where the cold trap usually persists, and changes to the zonal winds during this

year have a smaller effect because temperatures are more uniform. When the zonal winds are increased by 50% (panel d), the percent of trajectories experiencing their cold point between 120° E and 180° E increases to 37%, 60%, and 53% in 2007, 2008, and 2009, respectively.

It is notable that the cold point temperature and location is not significantly changed by the vertical wind speed in any scenario, including when the zonal wind is decelerated (one could expect these effects to amplify each other). This could be an artifact of our choice to apply a zonal mean acceleration to the BDC, rather than a scaling factor, which results in a zonal structure to the BDC's fractional change. In other words, the vertical velocity in regions where upwelling is already strongest will increase by less than our intended fractional changes of 20% or 50%. For example, the meridional and time mean (10° S and 10° N; January 20-29, 2008) upwelling at 100 hPa is as high as -0.01 Pa s$^{-1}$ in some small regions, while the zonal mean is -0.0007 Pa s$^{-1}$. This means that increasing the vertical velocity everywhere by 50% of the zonal mean (-0.00035 Pa s$^{-1}$) only increases the strongest background vertical velocity by 3.5%. Outside of that region, and especially above 90 hPa, the addition of -0.00035 Pa s$^{-1}$ is equal to a 10 to 20% acceleration or more. For the 2008 trajectories, the mean lifetime in the TTL (defined here as the time before crossing under 340 K for the first time) decreases from 62.3 d to 54.2 d (13%) when the zonal winds are increased by 50%, while the zonal distance traveled within the TTL decreases by 9%. Due to the localized nature of upwelling, our vertical wind scaling does not equate to a TTL lifetime scaling, and the TTL lifetime scaling does not equate to a zonal distance scaling. As mentioned in Section 2.2, Fueglistaler et al. (2014) did not find a difference between results when using 50% scaled and offset approaches, so we consider our result robust.

## 3.3 Temperatures histories near the cold point

Figure 5 shows the mean temperature of trajectories 24 hours before and after the cold point. By averaging over thousands of trajectories, random or periodic fluctuations in the temperature field cancel out, so Fig. 5 only shows the rate at which trajectories move through the average temperature field near the cold point. As is discussed in Section 3.4, these changes to the trajectory temperature history have potential implications for the life cycle of ice particles in the cold trap.

As one could expect, the rate of change of the trajectories' temperatures depends on the speed of the winds moving the trajectories, with faster winds resulting in more rapid temperature change. Figure 5d shows that the temperatures of the trajectories with both winds accelerated by 50% increase by 8.8 K within a day of experiencing their cold point, while the temperatures of the trajectories with unaltered winds only increase by about 7.2 K. Conversely, the increase in trajectories' temperatures within 24 hours of the cold point decreases to 5.2 K when the zonal winds are decelerated by 50%. This is due, in part, to changes to the residence time of trajectories within the cold trap as determined by the zonal wind speed.

The shape of these curves also depends on the temperature field that the trajectories are moving through, which will change as the location of the cold point changes. As was shown in Section 3.2, the deceleration of zonal winds by 50% decreases the number of trajectories that experience their cold point within the cold trap region from 57% to 30% in 2008 (note the shift in weight from 120° E–180° E to 270–330° E in Fig. 4c). Given that the cold trap has the strongest zonal temperature gradient within the TTL (see Fig. 1a), this shift means that 27% of the trajectories will experience their cold point within a weaker temperature gradient, causing a flattening of the curves in Fig. 5c beyond what is expected from changes to wind speed alone.

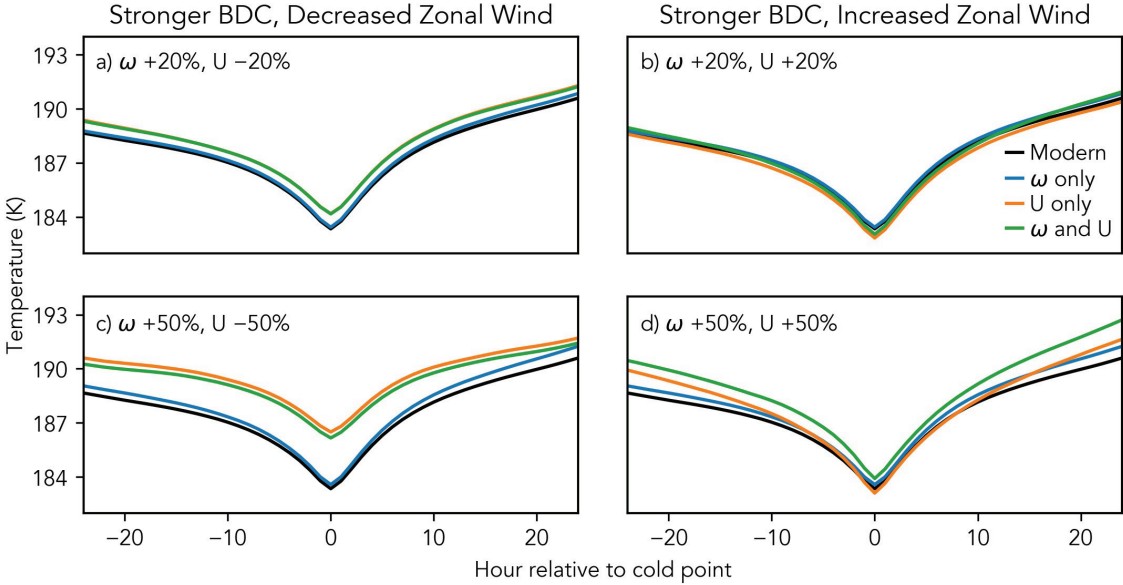

**Figure 5.** The mean temperature history of trajectories in the 24 hours before and after experiencing the cold point for DJF 2008 for the four altered transport scenarios: a) BDC accelerated by 20% and Walker Circulation decelerated by 20%, b) BDC accelerated by 20% and Walker Circulation accelerated by 20%, c) BDC accelerated by 50% and Walker Circulation decelerated by 50%, d) BDC accelerated by 50% and Walker Circulation accelerated by 50%. Figs. S7 and S8 show plots for DJF 2007 and 2009

## 3.4 Implications for stratospheric water vapor

To analyze the impacts of the trajectories' temperature histories described above, we next examine the life cycle of ice particles as the trajectories traverse the cold trap for our DJF 2008 trajectories. We assume that ice nucleation occurs at an ice saturation ratio of 1.6 (Koop et al., 2000), that sublimation subsequently occurs once an air parcel reaches an ice saturation ratio of 0.95 (Jensen et al., 2011), and that ice particle size distributions follow the observed values from Woods et al. (2018). We initialize trajectories with a water vapor concentration of 4 ppmv, which is consistent with background concentrations in the TTL during boreal winter. Reasonable changes to these parameters will change our exact results but not our conclusions. We also assume that the terminal velocity of ice particles can be calculated using a spherical geometry, which is reasonable given the particle habits observed at 180–190 K (Woods et al., 2018), and that ice particles exit the TTL after falling 1 km, which is both the approximate vertical thickness of the cold trap shown in Fig 1a and the median TTL cloud thickness observed by CALIOP during January 2009 (Schoeberl et al., 2014). See Fig. S9 for plots of the ice particle size distribution, fall speeds, and fraction of ice mass remaining in a 1 km layer following the cold point for the modern and 50% decelerated zonal winds scenarios.

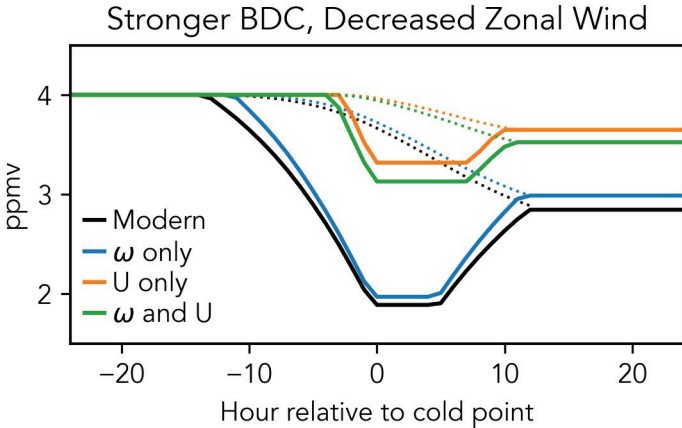

**Figure 6.** The water vapor concentration (solid lines) and the total water content (vapor + ice; dotted lines) for trajectories with a BDC accelerated by 50% and/or a Walker Circulation decelerated by 50% based on the temperatures in Fig. 5c, with ice nucleation occurring at an ice saturation ratio of 1.6, sublimation occurring at an ice saturation ratio of 0.95, and rehydration limited by ice sedimentation following the cold point.

### 3.4.1 Clausius-Clapeyron scaling

The most obvious and quantifiable impact of the altered temperature histories is from Clausius-Clapeyron scaling: the mean water vapor saturation mixing ratio calculated at each trajectory's cold point increases from 1.87 to 3.47 ppmv between the "modern" scenario and the 50% weakened Walker Circulation scenario. (See Table 2 for water vapor concentrations calculated
from the Clausius-Clapeyron equation for each scenario in 2008, and Tables A1 and A2 for 2007 and 2009.) If this calculation is done by first averaging the cold point temperatures of trajectories, rather than averaging the saturation mixing ratios of trajectories, the nonlinearity of the Clausius-Clapeyron relationship is ignored and the cold point water vapor mixing ratios are instead 1.89 ppmv and 3.32 ppmv. The difference in results from the two calculation methods shows the impact of the altered shape of the temperature distributions in Fig. 3c on water vapor concentrations. In Fig. 6, we calculate saturation mixing ratios
using the latter method for simplicity.

The water vapor concentration for the modern scenario reflects the known dry bias in water vapor reconstructions from Lagrangian trajectories (Liu et al., 2010), but water vapor anomalies have been shown to follow trajectory cold point variability (Fueglistaler et al., 2013; Dessler et al., 2014), so lower stratospheric water vapor would increase by about 1.60 ppmv with a 50% weakening of the zonal winds. For the more likely scenario of a 20% strengthening of the BDC and a 20% weakening of
the Walker Circulation, the mean saturation water vapor calculated from the trajectories' cold point temperature distribution increases by 0.34 ppmv to 2.21 ppmv (see Fig. 3a).

### 3.4.2 Nucleation, sublimation, and sedimentation

In addition to Clausius-Clapeyron scaling, the ice nucleation, sublimation, and sedimentation processes are impacted by the altered temperature histories near the cold point. These effects result in the differences in water vapor and total water content histories following the cold point in Fig 6, and they increase the final water vapor concentrations to 2.89 ppmv and 3.65 ppmv for the modern and 50% weakened Walker Circulation trajectories, respectively. This difference in rehydration reduces the net increase in the water vapor concentration that results from weakened zonal winds from 1.43 ppmv based on Clausius-Clapeyron scaling of average cold point temperatures to 0.76 ppmv.

The decreased rate of change of temperature following the cold point shifts the onset of sublimation from 5 hours after the cold point for the modern trajectories to 8 hours after for the 50% decreased zonal wind trajectories. This is shown graphically as an extension of the constant water vapor concentration following the cold point (hour 0) in Fig. 6. Based on observations from Woods et al. (2018), the increase in the cold point temperature from 183.4 K to 186.7 K also increases the mode particle diameter from approximately 18 $\mu$m to 21 $\mu$m, which increases the particle terminal fall speed by a factor of 1.3 and decreases the time required to fall 1 km from 20 hours to 15 hours (Heymsfield and Westbrook, 2010). This is reflected by the faster decrease of the total water content following nucleation for the decelerated trajectories in Fig. 6. These two effects – increased ice particle size and a more gradual increase in temperature following the cold point – increase the fraction of ice mass nucleated at the cold point that has fallen 1 km when sublimation begins from 11% to 31% when the zonal winds are decelerated by 50% (see Fig. S9d).

The ice mass at the cold point also depends on the mass of water vapor in the air approaching the cold point, and because the cold trap takes hours to transit, ice may nucleate prior to the cold point and be advected past it. Using our initial TTL water vapor concentration of 4 ppmv, we can calculate the ice mass that forms at each timestep prior to the cold point by calculating the decrease in the water vapor concentration relative to the previous timestep. We then add this to the remaining water vapor concentration to get the value that would result if all ice were sublimated, or the total water content. Ice particles will immediately begin to gravitationally settle, so we track the ice present at each timestep by summing up all ice formed prior to that timestep reduced by the fraction that has fallen 1 km. At the cold point, ice stops forming but it continues to sediment out, and once temperatures go above the sublimation threshold, ice is also converted to water vapor until total water content is equal to the water vapor concentration. At this point, there is no remaining ice and the water vapor concentration is fixed for the remainder of the timeseries. This procedure yields rehydrations of 0.96 ppmv and 0.33 ppmv following the cold point for the unaltered and 50% decelerated zonal wind trajectories, respectively.

Despite the large role for the life cycle of ice in the cold trap that we have illustrated here, the exact difference in rehydration between these scenarios depends on other factors, including the background water vapor and ice concentrations, availability of ice condensing nuclei, mixing, and radiative heating (Dinh et al., 2014). Additionally, gravity waves are ubiquitous in the TTL and can drive high frequency temperature fluctuations below 180 K (Fritts and Alexander, 2003). These waves have mainly been shown to decrease TTL water vapor concentrations by lowering the cold point temperature (Jensen and Pfister, 2004; Kim and Alexander, 2015), although this effect alone has been shown to only decrease water vapor concentrations by

about 0.2 ppmv (Wang et al., 2015). For waves with frequencies of greater than 2 cycles per day, which would be imposed on the temperature timeseries considered here, ice particles cannot sediment out before sublimating, and thus the waves have a negligible impact on final water vapor concentrations (Ueyama et al., 2015). Nonetheless, including waves and more detailed microphysics would complicate the picture of continuous nucleation, sedimentation, and sublimation that we have described here; as such, we emphasize that our result is meant to demonstrate a potential mechanism, not accurately quantify its impact.

## 4 Summary and Discussion

The dominant impact of zonal winds on future changes to the Lagrangian cold point shown in this work compliments the original cold trap hypothesis of Holton and Gettelman (2001). When the zonal winds are decreased by 50% for the DJF 2008 experiments shown in Section 3, the percent of trajectories that experience their coldest temperature between 120° E and 180° E drops from 57% to 30%, while the average cold point temperature increases from 183.4 K to 186.7 K. When considering the shifted temperature distribution without microphysical effects, this corresponds to a 1.6 ppmv increase in the water vapor concentration of air entering the stratosphere. This scenario is more extreme than what models and theory project though, so the warming of the temperature field will likely dominate stratospheric moistening, especially if there is a zonal warming pattern that weakens the importance of zonal winds. (In ACCMIP RCP 8.5 experiments, the January zonal mean cold point temperature increases by 1.7 K and the January mean cold trap temperature increases by 2.3 K, while the 20% decelerated zonal wind trajectories' Lagrangian cold point temperature increases by 0.8 K relative to that of the unaltered trajectories.) Nonetheless, future work concerning changes to stratospheric water vapor will need to consider changes to the Lagrangian cold point that will result from a weakened Walker Circulation.

The change in wind speed from the Walker Circulation deceleration will also impact the temperatures that air parcels and their suspended ice particles experience in the hours following their cold point. Decreased zonal wind speeds will delay the onset of sublimation of ice in regions with warmer temperatures, and the larger ice particles nucleated at warmer cold points will fall more rapidly out of the TTL. These effects decrease rehydration of air following the cold point and negate some of the moistening that results from increased cold point temperatures for decelerated trajectories. We estimate this rehydration reduction to compensate for over 40% of the increase in water vapor that would result solely from the warmer temperature histories of the trajectories with zonal winds weakened by 50%, but we ignore much of the complexity of the life cycle of ice particles in the TTL. Further work is needed to understand how changes to temperature, transport, and microphysics will combine to alter the dehydration efficiency of the cold trap in light of this work.

These results also present an important consideration for water vapor reconstructions done using trajectories that use lower temporal resolution input data. Previous work has shown that the water vapor concentrations calculated from trajectory cold points are lower than values observed in the lower stratosphere (Liu et al., 2010; Schoeberl and Dessler, 2011), with higher temporal resolution (1-h) trajectories being even colder and therefore drier than lower resolution (6-h) trajectories (Bourguet and Linz, 2022). One could therefore suppose that the 6-h trajectories are "more correct" than the 1-h trajectories due to their averaging over some characteristic timescale for microphysical processes. Our results show that this characteristic timescale

| Scenario | Cold point T (K) | % 120° E – 180° E | Cold point WV (ppmv) |
|---|---|---|---|
| Modern | 183.4 | 57 | 1.87 |
| Moderate BDC increase | 183.4 | 56 | 1.92 |
| **Strong BDC increase** | **183.6** | **56** | **1.98** |
| Moderate Walker Circulation increase | 182.8 | 63 | 1.73 |
| Moderate WC increase + Moderate BDC increase | 183.1 | 61 | 1.81 |
| Strong Walker Circulation increase* | 183.1 | 60 | 1.82 |
| Strong WC increase + Strong BDC increase* | 183.9 | 52 | 2.10 |
| **Moderate Walker Circulation decrease** | **184.2** | **46** | **2.19** |
| **Moderate WC decrease + Moderate BDC increase** | **184.2** | **47** | **2.21** |
| **Strong Walker Circulation decrease** | **186.7** | **30** | **3.47** |
| **Strong WC decrease + Strong BDC increase** | **186.4** | **32** | **3.28** |

**Table 2.** DJF 2008 trajectories' mean cold point temperatures and percentages within the cold trap (120° E – 180° E), and the saturation water vapor mixing ratio at an ice saturation ratio of 1.6 calculated from the cold point temperature distributions (WC = Walker Circulation). Bold text indicates warming greater than 0.2 K relative to the modern cold point temperature, with the exception of results impacted by excess dispersion, which are indicated with asterisks. Tables A1 and A2 show corresponding results for 2007 and 2009, respectively.

will change as transport in the TTL changes, and it is likely that trajectories run with 6-h input data are not able to capture this. Therefore, a robust method for calculating water vapor reconstructions requires input data with at least 1-h resolution and a consideration of the temperature histories in a time range spanning the cold point that is flexibly determined by the impact of temperatures on ice microphysics and sedimentation.

Although future changes may be subtle due to the gradual decrease in Walker Circulation strength and warming throughout the TTL, we have shown that the changes to the TTL's zonal wind field can change the location and temperature of the 345 Lagrangian cold point, as well as the time that air parcels spend in the vicinity of the cold point. This constitutes a stratospheric water vapor feedback that can emerge without changes to the TTL temperature field, and this feedback may act in concert with increased TTL temperatures to increase lower stratospheric water vapor beyond what is expected from Clausius-Clapeyron scaling. This can also lead to variability in the water vapor concentration that cannot be explained by changes to the temperature field, thereby adding complexity to analyses of lower stratospheric water vapor that rely on zonal mean or cold trap temperatures 350 alone.

*Code and data availability.* The ERA5 hourly data on native model levels used in this paper can be accessed through Copernicus Climate Change Service (C3S), https://apps.ecmwf.int/data-catalogues/era5/?class=ea. ACCMIP output can be obtained from NERC EDS Centre for Environmental Data Analysis, http://catalogue.ceda.ac.uk/uuid/ded523bf23d59910e5d73f1703a2d540. The source code for LAGRANTO powered by ERA5 data is available upon request from Michael Sprenger. Trajectory output is available upon request.

| Scenario | Cold point T (K) | % 120° E – 180° E | Cold point WV (ppmv) |
|---|---|---|---|
| Modern | 185.1 | 30 | 2.51 |
| Moderate BDC increase | 185.3 | 30 | 2.59 |
| **Strong BDC increase** | **185.6** | **30** | **2.74** |
| Moderate Walker Circulation increase | 185.2 | 35 | 2.59 |
| **Moderate WC increase + Moderate BDC increase** | **185.5** | **34** | **2.70** |
| Strong Walker Circulation increase[*] | 186.0 | 37 | 2.97 |
| Strong WC increase + Strong BDC increase[*] | 186.7 | 35 | 3.34 |
| Moderate Walker Circulation decrease | 185.3 | 26 | 2.62 |
| **Moderate WC decrease + Moderate BDC increase** | **185.4** | **27** | **2.66** |
| **Strong Walker Circulation decrease** | **187.0** | **23** | **3.57** |
| **Strong WC decrease + Strong BDC increase** | **186.6** | **26** | **3.33** |

**Table A1.** DJF 2007 trajectories' mean cold point temperatures and percentages within the cold trap (120° E – 180° E), and the saturation water vapor mixing ratio at an ice saturation ratio of 1.6 calculated from the cold point temperature distributions (WC = Walker Circulation). Bold text indicates warming greater than 0.2 K relative to the modern cold point temperature, with the exception of results impacted by excess dispersion, which are indicated with asterisks.

| Scenario | Cold point T (K) | % 120° E – 180° E | Cold point WV (ppmv) |
|---|---|---|---|
| Modern | 184.4 | 52 | 2.18 |
| Moderate BDC increase | 184.5 | 51 | 2.24 |
| **Strong BDC increase** | **184.7** | **49** | **2.36** |
| Moderate Walker Circulation increase | 184.4 | 53 | 2.21 |
| Moderate WC increase + Moderate BDC increase | 184.5 | 52 | 2.27 |
| Strong Walker Circulation increase[*] | 184.6 | 53 | 2.33 |
| Strong WC increase + Strong BDC increase[*] | 185.2 | 49 | 2.56 |
| **Moderate Walker Circulation decrease** | **184.7** | **44** | **2.33** |
| **Moderate WC decrease + Moderate BDC increase** | **184.5** | **46** | **2.28** |
| **Strong Walker Circulation decrease** | **186.5** | **27** | **3.29** |
| **Strong WC decrease + Strong BDC increase** | **186.2** | **30** | **3.15** |

**Table A2.** DJF 2009 trajectories' mean cold point temperatures and percentages within the cold trap (120° E – 180° E), and the saturation water vapor mixing ratio at an ice saturation ratio of 1.6 calculated from the cold point temperature distributions (WC = Walker Circulation). Bold text indicates warming greater than 0.2 K relative to the modern cold point temperature, with the exception of results impacted by excess dispersion, which are indicated with asterisks.

*Author contributions.* SB: conceptualization, data curation, formal analysis, investigation, methodology, visualization, writing– original draft preparation, review and editing. ML: refinement of ideas, resources, supervision, writing– review and editing

*Competing interests.* The authors have no competing interests to declare.

*Acknowledgements.* We thank Ed Gerber and one anonymous reviewer for constructive feedback that improved this manuscript. Thanks to Collen Golja, Ariana Castillo, and Boer Zhang for helpful feedback. ML acknowledges support from NASA New Investigator Program
Award 80NSSC21K0943.

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
