# Peer review of "Weakening of the Tropical Tropopause Layer Cold Trap with Global Warming"

_EGUsphere, 2023_

## Author Comment (AC1)

**Reviewer 1**

We thank Reviewer 1 for their feedback. We have improved our discussion of future changes to the TTL temperature structure and responded to your comments below.

This paper explores the impact of global warming on the TTL cold trap, which is expected to weaken due to the projected acceleration of the BDC and deceleration of the Walker circulation (WC). These expected changes in the future climate were proposed by Held and Soden (2006) or Vecchi and Soden (2007) and have been critically discussed in recent high-level publications (Lee et al., 2022, Chung at al., 2019, or Heede et al., 2021).

My main criticism of the paper is that it does not consider the anticipated changes in the temperature structure of the TTL in the future, which according to current knowledge (Brewer, 1949; Randel and Park, 2019), exerts a primary influence on the entry values of stratospheric water vapor (SWV). As indicated in the aforementioned papers, a future climate, resembling "El-Nino", is expected with a weaker cold trap over the Maritime Continent and a stronger cold trap over the Eastern Pacific. This is mainly due to the significant changes in the relative position and strength of the cold traps, i.e., their temperature minima, during El Ninos compared to La Ninas or neutral configurations, as our current understanding of stratospheric moistening due to strong El Ninos suggests (Scaife et al., 2003; Randel et al., 2009; Konopka et al., 2016). While I agree that the strength of the horizontal and vertical winds may play a role as a second-order effect, discussing the second-order effect without considering the first-order effect seems inappropriate to me.

This warming and its "El Nino" pattern are discussed in Section 2.3, where we show projections of TTL temperatures from ACCMIP's RCP 8.5 scenario. The January zonal mean warming within the TTL by the end of the century in these models is 1.7 K, while the warming in the cold trap region (120E to 180E) is 2.3 K. These numbers are 1.7 K and 2.1 K in the annual mean.

The trajectories run in 2008 with 20% and 50% decreased zonal wind speeds measure increases in the Lagrangian cold point temperature of 0.8 K and 3.3 K, so this effect is potentially of comparable importance to the overall warming of the TTL. This mechanism differs from modern variability, for which zonal mean temperatures are of primary importance, though in this hypothetical future with slower zonal wind speeds, interannual variability will still primarily be driven by zonal mean temperature variability. We are showing here that long-term trends in the background circulation are important for long-term trends in stratospheric water vapor, and we do not refute that temperatures within the TTL are the first-order explanation for the dryness of the stratosphere.

To address the concern of changes to the TTL's temperature structure, we have computed Lagrangian cold points temperatures and their associated water vapor concentrations with a cold trap (120E to 180E) warmed by 0.6 K. This temperature pattern resembles the asymmetric warming of the ACCMIP multi-model mean discussed above. (Additional zonal mean warming would offset all temperatures and therefore can be added in post processing to our Lagrangian cold point distributions.) Compared to the modern and 20% weakened Walker Circulation scenarios,

respectively, this results in an increase in the Lagrangian cold point temperature of 0.4 K and 0.3 K, and a decrease in the percent of trajectories that experience their cold point in the cold trap from 57% to 53% and 46% to 43%. Thus, the pattern of warming has little effect on the importance of the cold trap. This could also be expected by considering that the cold trap remains about 5 K colder than the rest of the TTL in the ACCMIP projections.

To address this concern in the text, we have added the following to Section 2.3:

*Quantitatively, there is a small amplification of warming in the cold trap, with increases of 1.7 K to the minimum zonal mean temperature and 2.3 K of the minimum mean between 120 E and 180 E (these numbers are 1.7 K and 2.1 K for the models' annual mean). This amplified warming within the cold trap will weaken the relevance of the cold trap hypothesis, but it does not eliminate the cold trap: the temperatures between 120 E and 180 E remain about 5 K colder than the zonal mean temperature. This amplification of cold trap warming has previously been linked to the deceleration of the WC (Hu et al., 2016), so our altered zonal wind scenarios would likely be impacted by patterns of temperature change similar to those shown in Fig. 2. To test this, we add 0.6 K to the temperature field between 120 E and 180 E for trajectories run with modern winds and with zonal winds reduced by 20% in 2008. By accounting for this asymmetric warming, the percent of trajectories that experience their cold point in this region from 57% to 51% and 46% to 41%, respectively. Thus, changes to the zonal wind speed dominate the response of the cold trap hypothesis to surface warming and our method is justified.*

Additionally, we add some discussion of modern ENSO variability in our results that is pertinent to your concerns.
In Section 3.1:
*This effect is smaller in 2007 and 2009, with increased mean cold point temperatures of 1.9 K and 2.1 K for trajectories with zonal wind speeds decelerated by 50\% in those years, respectively (Figs. S2 and S3). The variability across these years can be explained by the El Niño/Southern Oscillation (ENSO) state, which was weakly positive in 2007, weakly negative in 2008, and transitioning in early 2009. A positive ENSO index corresponds to a shifted and more diffuse cold trap, so changes to the zonal winds in the TTL should have less of an impact during El Niño years. This is an analog for the future warming pattern described in Section 2.3 -- El Niño-like warming may lessen but not eliminate the importance of zonal winds for TTL dehydration.*

And in Section 3.2:
*In 2009, the percent of trajectories experiencing their cold point between 120 E and 180 E decreases from 52% to 27% when zonal winds are decreased by 50%, while in 2007 the decrease is from 30% to 23% (Figs. S4 and S5). As was the case with the cold point temperatures, this is a signature of ENSO variability: the shifted and dispersed cold trap during the weak El Niño year (2007) causes fewer trajectories to experience their cold point within the region where the cold trap usually persists, and changes to the zonal winds during this year have a smaller effect because temperatures are more*

*uniform. When the zonal winds are increased by 50% (panel d), the percent of trajectories experiencing their cold point between 120 E and 180 E increases to 37%, 60%, and 53% in 2007, 2008, and 2009, respectively.*

I am curious why the authors did not use the meteorology provided by the ACCMIP. Using the trajectory-based method described in this paper for the 2000-09 meteorology and comparing it with the 2100-2109 data would be fantastic. First, separate the temperature effect, and then discuss the wind effect.

Thank you for this suggestion. Unfortunately, ACCMIP is not provided at hourly resolution (other than for surface pollutants). Previous work has shown the importance of using hourly data for studies of TTL dehydration (Bourguet and Linz, 2022), so our method optimizes model performance at the expense of more detailed wind and temperature changes. Nonetheless, our wind fields are supported by model projections and theory; thus, our study provides a simple mechanism that does not require climate model output.

Additionally, I recommend not including microphysics, which is only qualitatively discussed in this paper and, in my opinion, a third-order effect in the projected changes.

We appreciate the feedback, but choose to include this analysis because it reduces the primary effect discussed here. This analysis shows that the length of time that air spends near the cold point directly impacts dehydration efficiency. Previous work (Bourguet and Linz, 2022) has shown that microphysical effects are needed to match trajectory calculations to observations, and we show here that some of this microphysical effect will change following surface warming.

The authors' abstract claims to present a mechanism for a stratospheric water vapor feedback that can exist without changes to TTL temperatures. However, according to our current understanding, TTL temperatures are a result of wave forcing, which must change if an "El-Nino" like future is expected. Therefore, it is difficult to imagine changes in BDC and WC without changes in TTL temperatures. Overall, I find this study to be very idealized and recommend a "very major revision" or even withdrawal of the paper to explore the ideas discussed above. The only interesting statement in this paper is that "changes in the zonal wind are more important while changes to vertical winds have a negligible impact on water vapor concentrations." However, this statement in relation to vertical winds is even slightly different from the findings of Fueglistaler et al. (2014).

Our results with regard to the vertical winds agree with Fueglistaler et al. (2014), but our horizontal wind scaling is new. As mentioned above, we have now added a short analysis of TTL temperature changes and expanded our discussion of this effect.

Lower stratospheric water vapor is expected to increase with global warming, leading to a positive feedback on surface temperature. While much of this effect has been expected due to warming of the cold trap (the coldest temperatures in the TTL region, which set the amount of water vapor that can get through), the authors here explore the potential for changes in circulation to contribute to this positive feedback. In particular, they show that even if temperature structure remains fixed, the expected slowing of the Walker Circulation (and hence zonal winds in the TTL region) will also lead to a moistening of the stratosphere. This conclusion is obtained with some elegantly simple perturbation experiments, where zonal and vertical velocities obtained from ERA5 reanalysis are modified to capture the expected effects of greenhouse gas increase on the circulation. The key is that reduced zonal winds associated with a slowing of the Walker Circulation reduce the fraction of particles that reach the cold trap, effectively warming the Lagrangian cold point.

The authors also show that changes in circulation affect microphysical processes (sublimation and sedimentation), and ultimately moderate the potential drying of the air by the cold trap. While acknowledging the limitations of their experimental set up to fully capture all effects, they convincingly show that the full response cannot be properly simulated without a careful treatment of the circulation (in particular, its 3D spatial structure and short term temporal variability). This supports further investigation with more complex/high resolution models that capture both the dynamics and microphysics.

I recommend publication of this manuscript pending consideration of the minor concerns and suggestions listed below. I found the manuscript particularly well written and the figures well done, and commend the authors for the clarity of their exposition! It was a pleasure to read.

-Ed Gerber

Minor comments

(1) My only significant question about the manuscript concerns the divergent nature of the horizontal winds in the TTL. I was initially concerned that the resulting flow field will be inconsistent, with divergence in the zonal flow not properly balanced by meridional or vertical flow. Could this then lead particles to arbitrarily bunch up / disappear from regions with effective convergence/divergence? If so, could this bias your results if the coldest region is associated with divergence/convergence?

I know that the authors are aware of this (lines 110-115), as they explain this inconsistency is a problem because it can lead to artificial dispersion, particularly in the case of stronger winds. If dispersion leads to under sampling of the temperature field, I assume it would have a net

warming effect on the Lagrangian cold point, and thus if anything weaken the effects they see (where increased zonal wind cools the Lagrangian cold point).

To be constructive, I would have appreciated more discussion on this point. If previous studies have shown an inconsistency in the circulation to be minor when considering trajectories, or you have other reasons to believe that the effects are minor [say, that the distribution of particles stays uniform and my concerns above were unfounded], it would bolster confidence in the conclusions of the manuscript.

As is mentioned in the text, the meridional winds are much weaker than the zonal winds in the TTL, so scaling the meridional winds consistently should not make a large impact on the distribution of trajectories. In the "weakened Walker Circulation" scenarios, the small effect that weakened meridional winds would have is to slow the reverse trajectories' drift towards the equator, meaning that they should sample less meridional temperature variability as they cross the TTL. In other words, our "weakened Walker Circulation" trajectories (without meridional wind scaling) return to the center of divergence too quickly, but it is unclear what bias this introduces to the trajectories' temperature sampling.

Meridional motion near the center of divergence is very small, so vertical wind scaling is relatively important in mass conservation. Although our vertical wind scaling is based on a set fraction of the zonal mean rather than a fractional increase of the 4-D vertical wind field, the similarity between trajectories run with and without vertical wind scaling (the orange and green lines in our plots) suggests that the issue of "bunching" is not significant.

To add confidence to our assertion that the trajectory distributions are not subject to biased spatial sampling following our zonal wind scaling, I have plotted the trajectory distributions for 5 scenarios (modern winds, BDC +20%, BDC +50%, U -20%, and U -50%) after 40 days of integration. These are shown here and are now included in the supplemental. The exact distribution patterns differ between these plots, which is expected given the different wind fields, but there is qualitatively only more "bunching" when zonal winds are decreased by 50%. While a decrease in the meridional winds would likely decrease this effect, this is not problematic for our results. The meridional extent of the cold trap is large enough to encompass each of the distributions shown in this figure, so the impact of zonal sampling remains the primary driver of our results.

[Figure]

**Day 40 Trajectory Distributions**

a) Modern

b) BDC +20%

c) U -20%

d) BDC +50%

e) U -50%

Additionally, we have added to our discussion of dispersion in the context of the accelerated zonal wind trajectories at the end of Section 2.2:

*This scaled approach also creates excess dispersion in the accelerated zonal wind scenarios, which causes trajectories to undersample the spatial and temporal variability of the temperature*

*field. This effect does not impact our conclusions regarding the decelerated Walker Circulation, though it does obscure the decreased mean cold point temperature that results from the accelerated Walker Circulation, especially when zonal winds are increased by 50%. This dispersion decreases the amount of time spent in the TTL and the spatial sampling therein, so it introduces a warm bias that counteracts the decrease in the cold point temperature that would be associated with trajectories traversing greater zonal distances. Thus, our results for the accelerated zonal wind trajectories underestimate the effect of increased temperature sampling, but we do not attempt to correct for this given our focus on the decelerated zonal wind scenarios.*

(2) As purely a comment (not criticism), I am perplexed by the relative insensitivity of the Lagrangian cold point to vertical velocity omega, as opposed zonal wind u. Naively, I would have thought that omega would set the residence time in the TTL region, so that if you increase it, particles would have less time to sample the flow, and it would have the same impact as slowing the wind. I suspect my intuition is wrong because I am assuming the flow to be zonally uniform. The authors explain that variability in omega means that it barely changes in regions of maximum ascent/descent, even though the overall change is significant (20 or 50%). If most of the ascent is localized, changing the overall velocity may have little impact. (Or perhaps more subtly, it's about the location of the ascent relative to the cold point.)

To be constructive (and also stressing that these suggestions are minor: at the authors' discretion), could one provide more insight by considering how the parameter changes affect the total lifetime of particles in the TTL and/or their expected transit length. My naive expectation would have been that increasing/decreasing (in an absolute sense) omega would decrease/increase the time spent in the TTL, and so give particulars less/more time to roam and hence find the cold trap. But perhaps the localized nature of upwelling means this insight is wrong?

In terms of the zonal wind, naively I would expect a larger value of u to increase the mean distance particles travel (and hence increasing the temperatures they cover). Would this metric be easy to compute and confirm/dispell that intuition? It could be the case that even though the lifetime does change with omega, this has minimal effect on the distance particles travel due to zonal asymmetries.

Finally, perhaps even if my intuition was correct that changes in omega do change the lifetime of particles which does affect how far they can travel (faster BDC = less time to travel = less distance covered), the spatial structure of the flow doesn't really limit particles ability to see the cold trap in this case. In this limit, the careful trajectory calculations here are then the simplest way to capture the effects of omega and u on the Lagrangian cold point.

Defining the lifetime in the TTL is a bit tricky given that trajectories can oscillate around a cutoff (e.g. 340 K or 20 degrees N/S) before finally leaving the layer. For the trajectories with accelerated zonal winds, there is the added difficulty of their increased dispersion, which causes

some portion of them to leave the TTL too quickly. Therefore, the lifetime, distance traveled, and spatial sampling of trajectories with accelerated zonal winds are hard to interpret relative to the trajectories with unaltered zonal winds.

For simplicity, if we consider trajectories with accelerated vertical winds and consider the lifetime to be the amount of time before crossing 340 K for the first time, then the lifetimes of the modern, BDC +20%, and BDC +50% trajectories are 63.4, 60.1, and 55.3 days, respectively. The total zonal distances traveled by the BDC +20% and BDC +50% trajectories are 2% and 9% less than the modern trajectories. Thus, the fractional decrease in lifetime is greater than the decrease in zonal distance traveled, which highlights the localized nature of upwelling.

We have added a discussion of this in Section 3.2:
*For the 2008 trajectories, the mean lifetime in the TTL (defined here as the time before crossing under 340 K for the first time) decreases from 62.3 d to 54.2 d (13%) when the zonal winds are increased by 50%, while the zonal distance traveled within the TTL decreases by 9%. Thus, due to the localized nature of upwelling, our vertical wind scaling does not equate to a lifetime scaling, and the lifetime scaling does not equate to a zonal distance scaling.*

Despite the relatively weak response of trajectories to changes in vertical winds described here, the thermodynamic effect of increasing upwelling by these rates would be substantial. Therefore, we do not propose that vertical winds do not matter for air transiting the TTL, but only that changes to vertical winds do not impact the temperatures sampled as air parcels ascend to the lower stratosphere.

(3) There is little discussion of statistical significance in the body of the paper. I appreciate that the authors have computed these calculations from three different winters, but these figures are in the supplement. Could you include key results from these integrations to give a sense of the sampling uncertainty in the main text? For instance, at line 145 the impact on the 50% change of wind scenario is shown to increase the cold point from 183.4 to 186.7. Is this overall change identical in the other years — if so, you have high confidence. Or you could give the range of changes to give a rough sense of the sampling.

This suggestion was helpful and has improved our discussion of variability. I have added tables with the 2007 and 2009 results to the appendix, and I now reference these numbers in the text. The added comparison with 2007 (which was a weak El Nino year) was particularly useful and is included in Sections 3.1:
*This effect is smaller in 2007 and 2009, with increased mean cold point temperatures of 1.9 K and 2.1 K for trajectories with zonal wind speeds decelerated by 50\% in those years, respectively (Figs. S2 and S3). The variability across these years can be explained by the El Niño/Southern Oscillation (ENSO) state, which was weakly positive in 2007, weakly negative in 2008, and transitioning in early 2009. A positive ENSO index corresponds to a shifted and more diffuse cold trap, so changes to the zonal winds in the TTL should have less of an impact during*

*El Niño years. This is an analog for the future warming pattern described in Section 2.3 -- El Niño-like warming may lessen but not eliminate the importance of zonal winds for TTL dehydration.*

And in Section 3.2:
*In 2009, the percent of trajectories experiencing their cold point between 120 E and 180 E decreases from 52% to 27% when zonal winds are decreased by 50%, while in 2007 the decrease is from 30% to 23% (Figs. S4 and S5). As was the case with the cold point temperatures, this is a signature of ENSO variability: the shifted and dispersed cold trap during the weak El Niño year (2007) causes fewer trajectories to experience their cold point within the region where the cold trap usually persists, and changes to the zonal winds during this year have a smaller effect because temperatures are more uniform. When the zonal winds are increased by 50% (panel d), the percent of trajectories experiencing their cold point between 120 E and 180 E increases to 37%, 60%, and 53% in 2007, 2008, and 2009, respectively.*

Another option (which I think would also help sum up your results, independent of my concern about statistics) would be to include a summary figure that shows how the mean Lagrangian cold point varies with the wind perturbation.  Say, x axis = perturbation, -50% -20, 0 20 50, y axis Lagrangian cold point temperature.  Three lines could show how this varied across the three winters, with a different color to highlight what happened when the omega was changed or held fixed.

Thank you for this idea. We feel that results tables are more appropriate here given the different columns of data that we'd like to include. Additionally, we have a fairly small number of data points to include in this type of summary figure, and the U +50% data is difficult to interpret due to dispersion. Nonetheless, this comment has prompted us to reorganize the tables roughly in order of each experiment'schange to cold point temperature.

Very minor suggestions by line number

Is WC a standard acronym for the Walker Circulation?  Currently living in a country where that is more uniformly associated with a vital (albeit unrelated) facility, I found myself chuckling at times.  The text might flow more easily with Walker Circulation spelled out.  (This said, I am quite used to BDC for a shortening of the Brewer Dobson Circulation. This perhaps bias in the literature I know.)

We appreciate the delicacy with which you raise this concern. We've changed "WC" to Walker Circulation in almost all cases, other than in our summary tables which require abbreviated text.

19-20. Consider shifting "in 1949" to the start of the sentence, as I felt it broke up the connection between the water vapor observations and Alan's remarkable insight.

We've implemented this suggestion.

Figure 2. Consider adding headings above each column of panels with the model names, which would make the figure easier to follow quickly.

We've implemented this suggestion.

147 I think you don't have to "note" this fact. Rather, just state it, without ( ). "The trajectories with ...". In support of my comment (1) above, is it possible to quantify/show this dispersion effect?

We added to our discussion this dispersion effect throughout the paper, but quantifying it is beyond the scope of this work.

Figure 3 (like all the others) is well made, but I think more clear captions would make it easier to parse.
The top headings (+\omega, -U) could be put in words: "stronger BDC, decreased zonal wind" "strong BDC, increased zonal wind"
The inner captions could read a) \omega +20%, U -20%, b) \omega +20%, U +20\%, etc..
You could include information about the mean Lagrangian cold point with vertical bars that correspond with the colors of the different integrations. (These values are referred to in the text.)
Finally, would it be reasonable to include the basic Clausius-Clapeyron scaling on the WV changes in the other figures too?

These suggestions were very helpful, and the figure titles and captions have been changed accordingly.
We have decided to not add the vertical bars as you suggest because 1) the entire distribution needs to be considered when calculating water vapor concentrations, and highlighting the means in this figure would distract from the shifted distributions, and 2) the figure looks messy with the lines added.
We have added the Clausius-Clapeyron water vapor for panels b and c in Figure 3, but we omit it in panel d because 1) the trajectory dispersion when zonal winds are accelerated by 50% makes the water vapor calculation less meaningful, and 2) we would like to include the legend in that panel.

158 As noted in comment (3) above, is this tiny warming significant at all?

We've added the warming for 2007 and 2009, which is slightly larger but still small. It's hard to argue significance with a sample size of three, but insignificance would also support our point that the vertical wind changes shown are less important than the zonal wind changes.

Figure 6: I was initially confused how these curves appeared to be piecewise linear, at a temporal resolution longer than that of your model (1 hour). The text did eventually explain that these discrete jumps are a result of the highly nonlinear nature of ice nucleation and sublimation (starting at line 236). For my understanding, and perhaps for the readers, I assume these jumps would vary considerably for slightly different trajectories, so that if you were to do this analysis along the temperature structure of each individual trajectory, and then average, you would likely get a much smoother field. I trust this would be very nonlinear: do you expect that if you took into account variability, would the changes be amplified (more dehydration in the runs with weaker U) or weakened?

This is an interesting point. Due to the non-linearity of the Clausius-Clapeyron relationship and the non-linear dependencies of ice particle size on temperature and ice particle fall speed on ice particle size, I would expect this effect to be more pronounced when calculating trajectories separately and then averaging. There is an upper limit though – some trajectories may not condense any water vapor given that the starting value is already quite low (4 ppmv). I can see how adding these considerations could be insightful, but I think that this discussion would overcomplicate the point for most readers and distract from the conclusion.

260 There's a stray space: ') .'
Good catch, thanks.

Table 2 Could provide another opportunity to assess the statistical certainty, by contrasting results based on the other years? (Or do I misunderstand that you completed the full analysis on the other years. I do appreciate the immense effort it takes to do this analysis (hundreds of thousands of trajectories per experiment).
I have added tables for the other years in the appendix.

Alternatively, could you assess statistics from 2008 alone by sub-sampling trajectories and/or bootstrapping?
I prefer to do it for all 3 years in order to sample different TTL environments.

I want to emphasize that my concerns about the statistics are minor, in the sense that I feel this paper is more about identifying mechanisms and processes than trying to quantify trends vs. noise, where such quantification is essential. I do think the paper would be stronger, however, with more attention to uncertainties associated with sampling.
Thank you for raising concerns regarding statistical significance. Adding this significance has improved the paper.

---

## Author Response (AR2)

We thank our reviewer for constructive feedback. The manuscript has improved as a result of the minor revisions implemented following this review.

The paper has significantly improved, particularly addressing my main criticism regarding the assumption that Lagrangian trajectory analyses of future warming are only affected by changes in transport, without considering temperature changes (L150-160). Figure 2 and the related new discussion demonstrate that potential future temperature changes, as diagnosed in a few ACCMIP climate models (Lamarque et al., 2013), can be accounted for as an offset to all trajectories. This improvement enhances the consistency and justification of your procedure.

However, since this assumption holds significant importance, I would recommend mentioning it both in the abstract and in the conclusions.

For example, in the abstract:

"...Here, we investigate the impact of modified vertical and zonal wind velocities on the temperature histories of trajectories computed using ERA5 data, allowing us to examine the response of TTL transport during boreal winter to idealized changes in the BDC and Walker Circulation. Based on the findings of climate models, we assume that in Lagrangian trajectory analyses of future warming, changes in transport influence the trajectories, while temperature changes can be accounted for as an offset to all trajectories..."

We have added the following sentence to the abstract:
*Future changes to TTL temperatures can be applied as an offset to these temperature histories, including enhanced warming of the cold trap due to ``El Niño''-like warming, which has a secondary impact on the fraction of air that is dehydrated by the cold trap.*

The enhanced warming of the cold trap in ACCMIP projections is already discussed in the conclusion, so we opt to focus on our analysis of zonal wind changes and not add more on this temperature assumption in that section.

Lastly, I would like to note that in relevant publications such as Held and Soden (2006) or Vecchi and Soden (2007), the "El-Nino-like" future implies that the cold traps in the future over the Eastern Pacific should be more effective (i.e., colder) compared to the cold traps over the Western Pacific. Although the climate models used in Figure 2 do not support this expectation, I would recommend, not necessarily in this paper, to examine the newest versions of the models, such as those participating in CMIP6.

This is an interesting point. The cold temperatures over the Eastern Pacific do not increase as much as the cold temperatures over the Western Pacific in Figure 2, which should make the Eastern Pacific cold trap slightly more important for dehydration. The coldest temperatures

remain over the Western Pacific, but future work on the relative importance of these two cold traps could be important.

...and here some additional minor comments:

L22:
The seasonal and interannual variability of water vapor concentrations in the lower stratosphere can also be explained by the respective temperature variability in the TTL
Thank you for the suggestion. We prefer our current wording.

L26:
Surface climate is most sensitive to changes to water vapor concentrations near the tropopause (Forster and Shine, 1999; Solomon et al., 2010 and Riese et al., 2012 doi:10.1029/2012JD01775,1 2012), which are predicted to increase with
Thank you for the suggestion. We have added the citation.

L41:
This is crucial for explaining the dehydration of air entering the stratosphere, and it reconciles the incompatibility of the previous "stratospheric fountain" hypothesis with observations (Newell and Gould-Stewart, 1981).

I do not think there are any observations of net subsidence in coldest regions
We have added the citation for this (Sherwood 2000, *A stratospheric "drain" over the maritime continent*).

L78:
but is less impactful for input data with ERA5's resolution (Liu et al., 2010?, I think, you have to cite here other papers like Bourguet and Linz (2022) or maybe Legras and Bucci, 2020)
Thank you for the suggestion. We have added the citations as suggested.